# Evaluation of the Relationship between the BMI and the Sequence and Chronology of Eruption in Permanent Dentition in Spanish Population

**DOI:** 10.3390/healthcare10061046

**Published:** 2022-06-04

**Authors:** Marta Macarena Paz-Cortés, Laura Muñoz-Cano, Montserrat Diéguez-Pérez

**Affiliations:** 1Faculty of Dentistry, Alfonso X El Sabio University, Villanueva de la Cañada, 28691 Madrid, Spain; 2Faculty of Dentistry, European University of Madrid, Villaviciosa de Odón, 28670 Madrid, Spain; lau.mcano@gmail.com; 3Department of Preclinical Dentistry, European University of Madrid, Villaviciosa de Odón, 28670 Madrid, Spain; montserrat.dieguez@universidadeuropea.es

**Keywords:** pediatric dentistry, BMI, dental eruption, nutrition, obesity, overweight

## Abstract

The aim was to analyze the relationship between BMI (body mass index) and the sequence and chronology of the eruption of permanent teeth in a sample of Spanish children. Methods: The study design was descriptive, cross-sectional, observational, and epidemiological. Patients of pediatric age were included. Emerged teeth, and patient’s age, race, and sex were recorded. The nutritional status of the child was assessed by calculating the BMI, according to the WHO parameters. Statistical analysis was carried out with a confidence interval of 95%. A prediction model with logistic regression models was obtained. Results: A total of 725 pediatric patients between 4 and 14 years old were analyzed. BMI acts as a predictor variable for eruption symmetry, as it was most frequent in overweight children (*p* < 0.001). The probability of symmetry in dental eruption increases for South American children, for an extra month of age, and each meter of height. BMI had an influence in the first tooth to appear only in the fourth quadrant. BMI did not seem to influence present teeth, and the sequence of permanent dental eruption was not influenced by the BMI category. Conclusions: Age, weight, height, and BMI act as significant predictors for eruption symmetry. BMI does not produce alterations in the eruption sequence of the permanent dentition.

## 1. Introduction

Dental eruption is a dynamic process that involves the migration of the developing tooth from its intraosseous location to its final position in the oral cavity. Dental eruption encompasses the complete root development, the creation of the periodontium, and the establishment of a functional occlusion [1,2]. Although the clinical emergence of the tooth cannot be considered as complete dental eruption, they are traditionally considered synonyms [3]. Tooth eruption has been shown to be an orderly, sequential, and age-specific phenomenon [4,5]; beginning around 6 years of age and ending around 12 years of age [1,2,4,5], excluding third molars. Some racial variability in the age of tooth eruption and its sequence has been demonstrated [4,6]. The direct influence of variables such as genetics, hormonal factors, geographic location, race, sex, socioeconomic level, nutrition, and growth have been studied [6,7]. Recently, mutations in the PTHR1 gene have been associated with primary eruption failure (arrest of supraosseous eruption), and mutations in GNAS with other dental eruption defects [6,7]. Other factors, such as dentigerous cysts, and craniofacial deformities, such as cleft palate, have been described in association with abnormalities in dental eruption [8,9,10].

Nutritional status is the result of the requirement for different foods, and the intake, absorption, and use of the nutrients contained in the food [11]; affecting nutritional status affects growth and body development. Nutritional deficiency can retard growth, body size and proportions, chemical composition, and the quality and texture of certain tissues, such as bones and teeth [11,12]. The ADA has recognized the importance of nutrition and its direct influence on dental development. There are several methods for evaluating nutritional status in epidemiological studies, among which body mass index (BMI) [13] stands out, which allows individuals to be categorized as underweight or acutely malnourished, overweight or obese [1,14].

The WHO defines malnutrition as “the cellular imbalance between the supply of nutrients and energy and the demand of the body so that they can assure growth, maintenance, and specific functions”. Malnutrition also implies poor nutrition or deviations in the process, and is also related to breastfeeding and pregnancy [15,16]. Malnutrition remains the leading cause of death and the most common cause of poor health and poor growth in children and infants in developing countries [11,15]. Obesity is another nutritional problem that affects both adults and children and is considered the epidemic of the 21st century [17]. Childhood obesity has been related to multiple systemic pathologies, such as arterial hypertension, type II diabetes, dyslipidemia, obstructive sleep apnea, orthopedic and psychological problems, and alterations to the process of growth and development [12,17,18,19,20,21,22].

There are multiple studies that have related nutritional status and oral health [16,21,23,24,25,26,27,28,29], it even seems that alterations in the TAS2R38 gene (the bitter fret gene) are associated with caries rates and obesity [30]. Childhood overweight and obesity worsen aspects of oral health such as caries experience, periodontal disease, and quality of life in relation to oral health [31,32,33]. Changes in the shape, size, and dental eruption associated with lack of calcium and phosphorus during dental development have been reported [21,29,34]. Some associations between calcium, vitamin D, vitamin B, and ascorbic acid deficiency have also been shown with different forms of periodontal disease [11,12,14,15,17,20,22,23,24,25,26]. Alterations in the sequence of eruption and dental eruptive delay have been seen in premature children, and with low birth weight, endocrine disorders, race, or socioeconomic aspects; in addition to an increase in the incidence of malocclusions [12,25,26,27,28,29]. Recent studies suggested an advance in craniofacial growth in obese adolescents [21].

In the last decade, several studies have analyzed the influence of nutrition on dental eruption [6,11,12,15,17,20,21,26,29,34,35,36,37,38,39,40,41]. Most studies found an association between being overweight and acceleration of tooth eruption, although it was not significant in all cases [6,12,37,39,42]. Regarding malnutrition, there is no consensus, since while some authors established that there is a significant relationship [20], others found that there is no statistical significance [15]. The relationship between BMI and caries has been extensively studied [20,23,38,39,43]. A significant relationship was found between low weight due to malnutrition and delayed eruption of the permanent dentition, and the presence of untreated caries in primary dentition, in a longitudinal study in children in Indonesia [38]. However, there have been few studies at an international level, and few in the Spanish population, that have related BMI to the chronology and sequence of dental eruption. The aim of this study was to analyze the relationship between BMI and the sequence and chronology of the eruption of permanent teeth in a sample of Spanish children.

## 2. Materials and Methods

A descriptive and cross-sectional observational epidemiological study was carried out between November 2020 and September 2021, to determine the chronology and sequence of the eruption of the permanent dentition in a pediatric population of the Community of Madrid, Spain. The accessible population were randomly selected children who attended the Dental Clinic of the European University of Madrid and two private clinics in Madrid. For the calculation of the statistical power, the estimation of the age of eruption of the permanent dentition with a precision of 0.25 years and a confidence level of 95% was taken into account. Based on a standard deviation of 0.5 years in previous studies, a minimum of 16 patients were required in each defined age group for each sex, so the minimum number of patients to obtain statistical power was 620. Patients in pediatric age were included, excluding patients with systemic diseases, syndromes, local alterations of dental eruption such as dentigerous cysts, craniofacial deformities, general developmental disorders, and dental anomalies (including dental agenesis). This study complied with the Declaration of Helsinki, and had the approval of the Research Committee of the European University of Madrid and the Ethics Committee of the Community of Madrid (codes uem10122019 and CIPI/19/108). Prior to conducting the study, the parents or legal guardians of the patients were informed and signed an informed consent to be part of the study.

Emerged teeth were recorded by intraoral examination in the dental chair, with lighting and the help of an intraoral mirror. Clinical eruption was considered as the moment when the tooth penetrates the gingival mucosa and becomes clinically visible, according to Carr et al. [44]. Registration was carried out dichotomously (presence/absence) and third molars were excluded from the study. The patient’s age, race (according to both parental and grandparental race), and sex were also recorded. The nutritional status of the child was assessed by calculating the BMI (kg/m^2^) according to the WHO parameters [20,22] and the International Obesity Task Force (IOTF) [45], classifying the values as underweight (BMI < 18.5), normal weight (BMI 18.5–24.9), or overweight (BMI 25–29.9). A second examination was performed randomly on 20% of the sample one week after the baseline, by the principal investigator and the second observer, to perform an analysis of intra- and inter-observer agreement. Tables of the BMI of Spanish children were used [45].

Statistical analysis was carried out with the Statistical Software R (version 4.1.1) using the RStudio environment, using a confidence interval of 95% (statistical significance of 5%) for the contrast of hypotheses. Descriptive statistics were carried out using numerical summaries (mean and standard deviation) for quantitative variables, and frequency and contingency tables for qualitative variables. The study of the relationship between qualitative factors was carried out using the Chi-square test (χ^2^) or Fisher’s exact test, depending on the sample characteristics. Prediction models were utilized for one factor with logistic regression models, with the Wald test and the ROC and AUC curves. Whether the sample met normality criteria was studied with the Levene test, assuming normality when the sizes were large enough by the central limit theorem. The comparison of means in two groups was performed with the Student’s T-test (assuming the normality of the sample), Welch’s T-test (not assuming normality of the sample), and Wilcoxon’s test (very small sample size). The comparison of means in more than two groups was carried out with the ANOVA test (assuming normality of the sample), the Welch ANOVA test (in the event of a certain lack of normality), or the Kruskal–Wallis test with the Wilcoxon comparisons (lack of clearly normal). The relationship between the numerical variables was studied with the correlation coefficient and Spearman’s correlation test.

## 3. Results

A pediatric sample of 725 patients (367 girls, 358 boys) aged between 4 and 14 years was analyzed. In relation to race description, 93.1% of the sample was Caucasian and the remaining 6.9% South Americans. Regarding the weight category, only 47.72% of the sample presented normal weight, while 49.52% presented low weight and 2.76% were overweight. The analysis of intra- and inter-operator agreement was carried out; obtaining Kappa values equal to 1 (perfect agreement) in the intra-operator evaluation, and greater than 0.95 (perfect or almost perfect agreement) in the inter-operator analysis.

The relationship between the sex, age, race, BMI, and BMI category of the subjects was analyzed (Table 1, Figure 1 and Figure 2), without finding a significant association between BMI classification, sex, or race. The mean age (115 months) was the same in boys and girls. Significant differences were found in the mean age in relation to BMI classification and race, since the mean age was higher in the group of overweight children and in the group of Caucasian children. The BMI analysis did not present significant differences with respect to sex or race. The correlation between the numerical variables was analyzed, finding a very strong positive and significant correlation between age and BMI, with a correlation coefficient of 0.549 (*p* Value < 0.001) (Figure 3).

The symmetry in dental eruption between hemiarches was analyzed, finding symmetry in only 41.38% of the sample, with statistically significant differences (*p* Value < 0.001) with respect to the group with asymmetry (68.62%). The mean values of age and BMI were higher in the group with symmetry (*p* < 0.001) (Figure 4). The relationship between superior and inferior symmetry was analyzed, with their association being relevant, since superior symmetry was also found in 67.87% of the cases with inferior symmetry (χ^2^
*p* Value < 0.001). Regarding the influence of sex, race, and BMI category, statistically significant differences were only found in the BMI category (χ^2^
*p* Value < 0.001); eruption symmetry was present in 28.41% of underweight children, 52.89% of normal weight children, and 75% of overweight children. According to the univariate logistic regression model, both BMI category and BMI and age act as significant predictors of eruption symmetry (*p* Value < 0.001 in all analyses), since the OR indicates an increasing relationship with symmetry. A multivariate logistic regression model was performed, showing that race, age, and height are predictors of eruption symmetry (Wald *p* Value = 0). The probability of eruption symmetry was multiplied by 2.5 for South American individuals, multiplied by 1.013 for each extra month of age, and multiplied by 866.75 for each meter of height increase (Figure 5).

It was found that the first tooth to appear in the maxilla was the first molar (74.34–76.41% of the sample), while in the mandible it was the central incisor (78.9–82.76% of the sample), with statistically significant differences (χ^2^
*p* Value < 0.001). The relationship between eruption symmetry and the first tooth to appear was analyzed, finding that, in the fourth quadrant, if the first molar was the first tooth to erupt, eruption asymmetry was present in 80.95% of cases (Fisher *p* Value = 0.003391). Furthermore, in the maxilla, the first molar was the first tooth to appear in slower developing children, as opposed to the mandibular incisor (Wilcoxon *p* Value < 0.05 in all comparisons). In quadrants 1 and 3, no relationship was found with any of the factors studied, while the first molar was by far the first tooth in Caucasian patients in the second quartile (Fisher *p* Value = 0.01365), and the first molar erupted the first in males in the fourth quadrant (Fisher *p* Value = 0.04563).

According to the univariate logistic regression model, age, weight, and height have a significant influence on whether the first permanent molar is the first tooth to appear in all quadrants, while BMI only has an influence in the fourth quadrant. The multivariate logistic regression model revealed that only the first tooth to erupt in the fourth quadrant could be predicted; therefore, it does not make sense to propose a predictive model for the first tooth to erupt. The probability that the first tooth would be the first molar in the fourth quadrant was 3.359 times higher for males than for females (Wald *p* Value = 0.021).

The behavior of the teeth present in relation to age and BMI category was studied, finding that age was significant in all cases (*p* Wald < 0.05 in all comparisons) (the older the teeth, the more likely they were to be erupted), while BMI category or sex did not seem to influence tooth eruption. The sequence of eruption was analyzed, determining that the order of eruption of the teeth was not the same in all the quadrants, and that it was not influenced by the BMI category (Table 2, Figure 6).

## 4. Discussion

The present study was cross-sectional, analyzing 725 Spanish children of Caucasian and South American race between 4 and 14 years of age; where in 93.1% of the cases they were Caucasian children, so the data referring to race differences should be interpreted with caution. In the present study, dental emergency was determined by orthopantomography and clinical visual examination, classifying the BMI using the WHO and the International Obesity Task Force (IOTF) criteria [16,45]. The choice of these criteria was due to the fact that they are the most used for the classification of childhood obesity, and therefore, for the performance of an adequate comparison and discussion between articles. Thus, since the sample was of Spanish children, it was decided to use BMI tables in the Spanish population, so that they would be representative. In the analysis of the previous literature on the relationship between BMI and the chronology and eruption sequence of permanent teeth, both longitudinal [17,36,38] and cross-sectional [2,4,6,20,21,37,39,46,47,48,49] studies were found. With regard to race, most previous studies were in Asian children [4,14,39,47,48], although there are also some in North American children [20,35], Spanish [2,42,46], Central Americans [17], Africans [6], Europeans [37], and South Americans [11,21,49]. Regarding the sample size and the age of the children in the existing literature, this is very heterogeneous, ranging between 50 and 3519 participants, and between birth and 19 years of age, with different age ranges between the studies [2,6,17,20,21,35,36,37,42,46,47,48,50].

There are several studies that analyzed weight and height in relation to the eruption of permanent teeth. A study carried out in children concluded that pediatric patients with a genetic growth deficit did not present delays in the eruption of the permanent dentition, while those with growth hormone deficiency had a significant delay in dental turnover; however, the data in relation to the weight of the subjects were not significant [46]. In an investigation with Spanish children, a positive relationship was obtained with both weight and height and the eruption of the permanent dentition for both sexes and for all age groups [2]. In Mexican children, a positive and direct relationship was also found between weight and height with dental eruption, with a positive correlation with weight in 46% of girls and 64% of boys, and a positive correlation with height in 48% of girls and 68% of boys [50]. Gaur et al. also found a significant correlation between weight and height with the eruption of the permanent dentition; however, when the age factor was eliminated from the analysis, these differences were not statistically significant [50]. Kutesa et al. [6] did not find a relationship between the chronology of eruption and height, but they did find a relationship with weight, since it was correlated with the mean time of eruption in half of the teeth analyzed.

Regarding BMI, there are several studies that considered the differences with respect to the chronology of eruption. In our results, in agreement with the results of previous authors, despite finding a strong association between height and weight and the eruption of the permanent dentition, BMI category did not behave as a predictor variable for the appearance of the permanent teeth in any of the cases. In North American children, overweight or obese pediatric patients were found to have accelerated dental eruption [20]. Sánchez-Pérez et al. [17] concluded that being overweight significantly accelerates the eruption of permanent teeth, since overweight children had an average of five more teeth than thin children. A relevant study carried out in the USA determined that obese children have a greater number of erupted permanent teeth during the mixed dentition, presenting 1.44 more teeth than non-obese children, regardless of their sex, age, and race [35]; the difference being more pronounced in children aged 10–11 years in relation to puberty. Similar results were found in other studies, which specified that children with a high BMI would have 1.9 more permanent teeth erupted than children with a low BMI, and one tooth more than those with normal weight [48]. In a recent study carried out on Spanish children, they found that being overweight and obesity increase the probability of early eruption of permanent teeth by 1.5–2 compared to normal-weight children. In addition, as in our study, they found a strong positive relationship between age and the probability of dental eruption, finding that the impact of obesity on dental eruption remains stable for the different age ranges studied [42]. Arid et al. analyzed the percentage of patients with eruptive delay, finding an agreement with previous studies that 100% of underweight children had a significant eruptive delay, compared to 32.5% in overweight children and a 16.67% in obese children [11]. A high BMI acts as a predictive factor for an earlier eruption, since a generalized earlier eruption of permanent teeth has been shown [37]; and was also found regardless of the subject’s sex [21], which allowed the elaboration of a regression model, whereby we can use age, BMI, and sex to predict the number of erupted teeth [21,47].

It has been determined by longitudinal analysis of a cohort of children that both being overweight and obesity cause early eruption of permanent teeth, since a high BMI at young ages predisposes to accelerated dental eruption, which persists throughout growth [28]. Low weight or malnutrition is associated with a lower number of erupted teeth in children; in addition, low weight or malnutrition at the age of 6–7 years is related to fewer permanent teeth having erupted at 8–9 years [38].

On the other hand, in children from India, they found an advance in dental eruption of the permanent dentition in relation to height, weight, and advanced sexual maturity, but not in relation to BMI [4]. In addition, the study by Anu et al. established the presence of a significant negative correlation between BMI and lower central incisors, finding an eruptive delay of the aforementioned teeth in obese children [39], contrary to reports by other authors.

The analyzed articles did not study the relationship between BMI and tooth eruption symmetry. Our results indicate that age, height, weight, BMI, and BMI category behave as predictive variables of eruption symmetry, allowing the establishment of a regression model between height and eruption symmetry. In the study by Hernández et al. [2], they analyzed the symmetry of eruption, establishing that there were no statistically significant differences between dental eruption in the right and left hemiarch. Gaur et al. [50], on the other hand, found that there were only significant differences in the eruption of the maxillary central incisors and mandibular canines between the two hemiarches in children. A recent systematic review [51] analyzed the genetic factors associated with asymmetric mandibular growth, finding potential etiological factors including PITX2, ACTN3, ENPP1, and ESR1. However, to date the scientific evidence of this association is scarce; and, furthermore, no significant relationship has been found between asymmetric mandibular growth and asymmetry in dental eruption.

No previous studies were found in relation to the association between the first tooth to appear and weight, height, or BMI. According to our results, the first tooth to erupt, age, height, and weight have a significant influence on whether the first permanent molar is the first tooth to erupt in all four quadrants, acting as predictor variables. BMI acts as a predictor variable for the first molar being the first tooth to erupt only in the fourth quadrant, while BMI category does not present a significant association. Our logistic regression model did not present consistent results in the four quadrants with respect to the first tooth to erupt and variables such as sex, race, or BMI, so we concluded that the creation of the mathematical model was not possible. Previous studies affirmed that the first tooth to erupt in the maxilla is the first permanent molar and, in the mandible, the central incisor or the first permanent molar [2,4,6,21,37,47,50]; although they did not evaluate the relationship between this and height, weight, or BMI.

Our data establish that the eruption sequence does not vary depending on the BMI category. In a study carried out in [47] Peruvian children between 3 and 12 years of age, they analyzed whether BMI affected the dental eruption sequence, following the Logan-Kronfeld classification, finding that 23% of malnourished children presented an altered eruption sequence, although without statistical significance [49]. In previous studies, it was described that the sequence of eruption in the mandible is 6-1-2-3-4-5-7 and in the maxilla 6-1-2-4-3-5-7, although they did not make comparisons with height, weight, or BMI [4].

We currently find that childhood obesity is a growing problem among the pediatric population. The influence of BMI on the sequence and timing of the permanent dentition has great clinical relevance for dental treatment planning. The chronology and sequence of the eruption of the permanent dentition is basic in orthodontic treatment, and for the determination of chronological age from dental age in forensic medicine [20,34,35,36,47]. Furthermore, from a biological point of view, it is vitally important to integrate bodily development into the oral health of our patients.

The present study has some limitations. Since the exclusion criteria eliminated patients with craniofacial alterations or dental anomalies (agenesia), the difference between these and conventional children cannot be determined. In addition, only articles published in English or Spanish were chosen for discussion, so there may be a small publication bias. Weight and height, and therefore the BMI, are variable anthropometric criteria in which the socioeconomic level, race, diet, and evolutionary changes have repercussions; thus, the comparison with research carried out in other geographical locations or in previous decades may be subject to errors. Since the values and classification criteria of BMI are variable, it is important to bear in mind that those used for the present study correspond to those in force in the Spanish population and at the date of publication, so caution should be exercised when interpreting these data in other populations or races. It would be of great interest to be able to carry out a longitudinal study by increasing the sample size, in order to avoid possible biases. Due to the growing prevalence of obesity in the general population at an international level, and increasing in the child population, we consider the relationship between nutritional level and dentofacial characteristics of great relevance. The prediction of the eruption of the permanent dentition is of vital importance in the planning of not only dental, but also orthodontic/orthopedic, treatments.

## 5. Conclusions

Eruption symmetry in the hemiarches occurred in less than 50% of subjects, with age, weight, height, and BMI acting as significant predictors for symmetry. There was a significant relationship between the existence of symmetry in the upper and lower jaw. The first permanent molar was the first tooth to erupt in the upper jaw, and the central incisor in the lower jaw. It was not possible to establish a predictive model for the eruption of the first tooth to erupt. BMI did not produce alterations in the eruption sequence of the permanent dentition.

## Figures and Tables

**Figure 1 healthcare-10-01046-f001:**
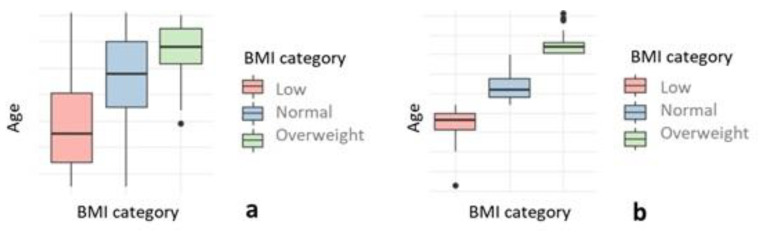
Box plot of age (**a**), and BMI (**b**), according to BMI category.

**Figure 2 healthcare-10-01046-f002:**
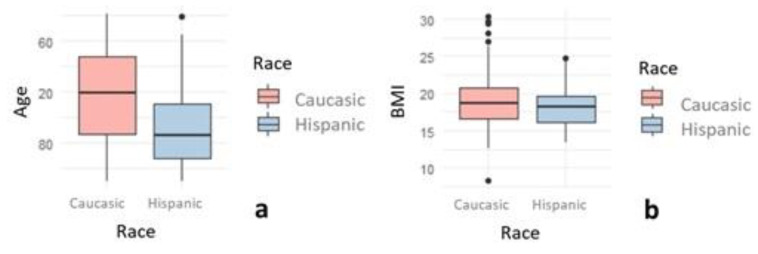
Box plot of age (**a**), and BMI (**b**), according to race.

**Figure 3 healthcare-10-01046-f003:**
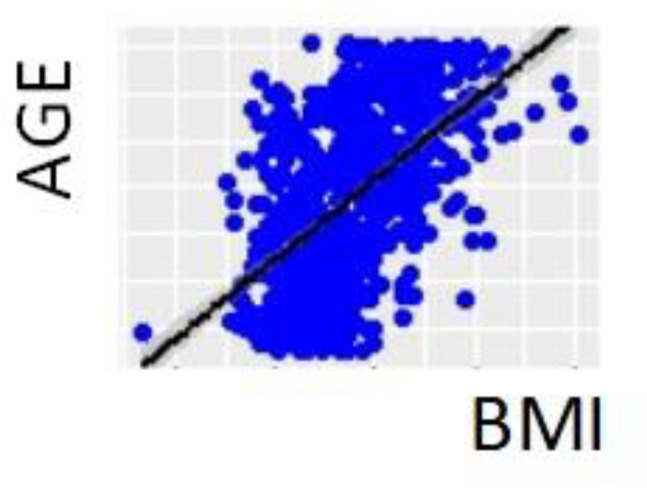
Dispersion diagram of age and BMI. Positive strong correlation between age and BMI. Correlation coefficient 0.549 (*p* Value < 0.001).

**Figure 4 healthcare-10-01046-f004:**
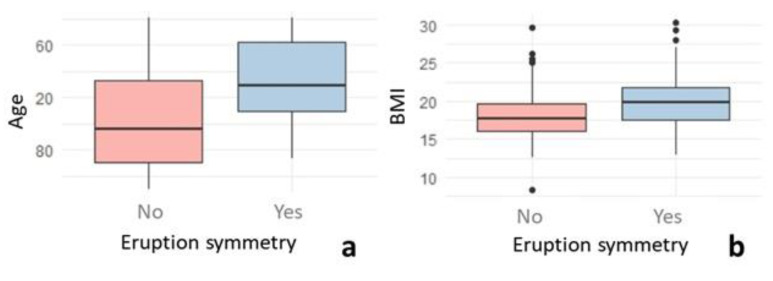
Box plot of age (**a**) and BMI (**b**) as a function of eruption symmetry.

**Figure 5 healthcare-10-01046-f005:**
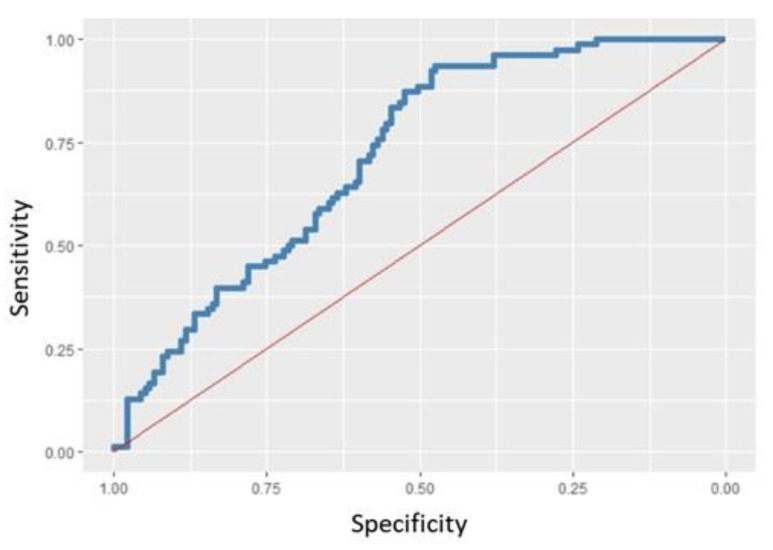
ROC curve for the multivariate logistic regression model with AUC = 0.7181 for eruption symmetry. 70% of the sample was used as training and 30% as testing, obtaining a correct classification of 67.28% of the individuals.

**Figure 6 healthcare-10-01046-f006:**
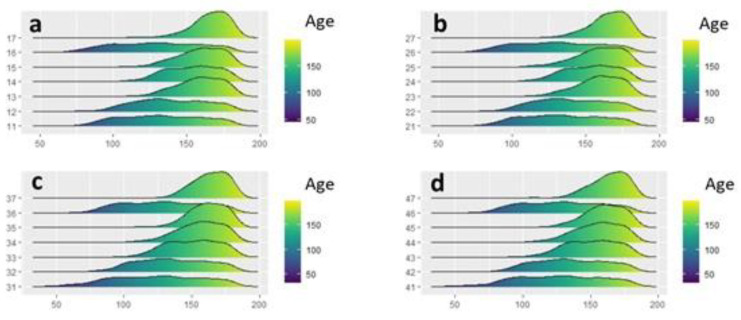
Estimation of the density of the age of the individuals having each tooth by quadrant: first quadrant (**a**), second quadrant (**b**), third quadrant (**c**), fourth quadrant (**d**).

**Table 1 healthcare-10-01046-t001:** Relationship between the variables BMI category, race, sex, age, height, weight, and BMI.

		BMICat. ^#^			Race ^¬^		Age (Months)		BMI (kg/m^2^)	
		1	2	3	C	H	Mean	SD ^+^	Mean	SD
**Sex ***	**Sig.**	*p*-ꭓ^2^ = 0.6582			*p*-ꭓ^2^ = 1		*p*-Ttest = 0.9159		*p*-Ttest = 0.6262	
	**F**	183	172	12	342	25	115	36.7	18.8	3.02
	**M**	176	174	8	333	25	115	37.2	18.7	2.95
**IMC cat.**	**Sig.**				*p*-ꭓ^2^ = 0.3511		*p*-Welch < 0.001		*p*-Welch < 0.001	
	**1**				331	28	96.4	31.5	16.3	1.42
	**2**				324	22	133	32.5	20.8	1.64
	**3**				20	0	153	21.6	26.4	1.64
**Race**	**Sig.**						*p*-Welch < 0.001		*p*-Ttest = 0.1536	
	**C**						117	36.8	18.8	3.00
	**H**						91.9	31.1	18.2	2.61

* Sex. F (female), M (male). ^#^ BMC Category. 1 (underweight), 2 (norm weight), 3 (overweight). ^+^ SD. Standard Deviation. ^¬^ Race. C (Caucasian), H (South Americans).

**Table 2 healthcare-10-01046-t002:** Chronology of dental eruption by quadrant. Tooth 1 (central incisor), tooth 2 (lateral incisor), tooth 3 (canine), tooth 4 (first premolar), tooth 5 (second premolar), tooth 6 (first molar), tooth 7 (second premolar).

Quadrant	Eruption Order
First quadrant	1 ≡ 6 < 2 < 4 < 3 ≡ 5 < 7
Second quadrant	
Third quadrant	1 < 6 < 2 < 3 < 4 < 5 < 7
Fourth quadrant	1 ≡ 6 < 2 < 3 < 4 < 5 < 7

## Data Availability

The data presented in this study are available on request from the corresponding author (M.M.P.-C.). The data are not publicly available, due to the disagreement of the participants.

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
