# Peer review of "Evaluation of the Relationship between the BMI and the Sequence and Chronology of Eruption in Permanent Dentition in Spanish Population"

_healthcare, 2022, doi:10.3390/healthcare10061046_

Round 1

Reviewer 1 Report

Dear Editor,

thank you for asking me to serve as reviewer for this article. Please find below my comments.

Abstract: BMI is used without explanation. The first time it should be included to what the acronym corresponds

Introduction and manuscript: Authors seem to overlap the concepts of “dental eruption” with “dental development”, whilst the development is the calcification of crown-roots of teeth. For instance a completely developed tooth can have a delayed eruption or be impacted.

Material an method: the ancestry of the children should be included. Both BMI and eruption time could be influnced by ancestry. Ethnicity and race are not equivalent terms. Moreover  auhtors should explain what they intend with Hispanics compared to Caucasics. According to my knowledge Hispanics are people that speak Spanish and the term “Hispanic” are used in US for indicating people coming from Mexico or Argentina (speaking Spanish). Since this paper considers Spanish children it is not clear to what ancestry the two considered racial groups refer to.

I suppose that sex (biological sex) of individuals was considered and not the gender.

Generally speaking the article is endowed by less  novelty and gains low scientific interest. I suggest to Authors to clearly indicate the rationale behind the study. In my opinion the fact that a possible correlation between two variables (BMI and eruption in thsi case) has not been studied in a certain population makes the study interesting. Maybe a they could indicate the clinical (or whatever) relevance of such a correlation, if any.  

Author Response

Dear Reviewer #1. Thank you very much for your time and dedication to correcting our manuscript. We hope to be able to respond to your doubts adequately and to improve the quality of our manuscript with your considerations.

It is true that in the main text the BMI acronym was explained, but not in the abstract. It is added to the abstract, and the formula is specified in material and methods.

Thank you very much for your consideration in relation to “dental eruption and development”, which we find correct. In some sentences the concepts of “dental eruption” and “dental development” were wrong. The manuscript has been reviewed and corrected.

Regarding the methodology of the study, the ancestors of the children were not specified in the study, but it was taken into consideration that both the parents and the grandparents of the children were of the same race for their classification, avoiding biases in the variables to be measured in the study. This consideration is added in material and methods.

Grammatical considerations:

  1. We proceed to change Hispanic to South Americans. The predominant race in Spain was Caucasian, but due to the high prevalence of immigration, it was decided to include race as a variable to avoid bias.
  2. Change ethnicity or ethnicity to race.
  3. We change gender for sex

Due to the growing prevalence of obesity in the general population at the international level, and growing in the child population, we consider that the relationship between nutritional level and dentofacial characteristics is of great relevance. The prediction of the eruption of the permanent dentition is of vital importance in the planning of not only dental but orthodontic/orthopedic treatments. In addition, the scant literature on this topic in the Spanish pediatric population strengthens the importance of our study.

Reviewer 2 Report

Dear Sirs, thank you for opportunity to revew this interesting article. I would like to point out some flaws though:

  1. The references are pretty old (I counted 15 out of 40 that are form the past 10 years)
  2. In materials and methods please, add the graph(s) how does BMI combine with boys and girls
  3. It would be interesting to add some points that may influence the tooth eruption, regardless BMI like - cysts Nahajowski M, Hnitecka S, Antoszewska-Smith J, Rumin K, Dubowik M, Sarul M. Factors influencing an eruption of teeth associated with a dentigerous cyst: a systematic review and meta-analysis. BMC Oral Health. 2021 Apr 7;21(1):180. doi: 10.1186/s12903-021-01542-y.; genetics / facial deformities/ clefts Paradowska-Stolarz A. MSX1 gene in the etiology orofacial deformities. Postepy Hig Med Dosw (Online). 2015 Dec 31;69:1499-504. PMID: 27259221.; Almotairy N, Pegelow M. Dental age comparison in patients born with unilateral cleft lip and palate to a control sample using Demirjian and Willems methods. Eur J Orthod. 2018 Jan 23;40(1):74-81. doi: 10.1093/ejo/cjx031.; 
  4. Please, add genetic basis in the introduction: Kiliç M, Gurbuz T, Kahraman CY, Cayir A, Bilgiç A, Kurt Y. Relationship between the TAS2R38 and TAS1R2 polymorphisms and the dental status in obese children [published online as ahead of print on May 4, 2022]. Dent Med Probl. doi:10.17219/dmp/143252
  5. Please, refer to dental status of obese and overweight patients in the introduction: DeszczyÅ„ska K, Górska R, HaÅ‚adyj A. Clinical condition of the oral cavity in overweight and obese patients. Dent Med Probl. 2021;58(2):147–154. doi:10.17219/dmp/127873
  6. Please, refer more to contributions in food intake and the tooth eruption pattern: Dimaisip-Nabuab J, Duijster D, Benzian H, Heinrich-Weltzien R, Homsavath A, Monse B, Sithan H, Stauf N, Susilawati S, Kromeyer-Hauschild K. Nutritional status, dental caries and tooth eruption in children: a longitudinal study in Cambodia, Indonesia and Lao PDR. BMC Pediatr. 2018 Sep 14;18(1):300. doi: 10.1186/s12887-018-1277-6. PMID: 30217185; PMCID: PMC6137874.
  7. Please, rescan the materials, methods etc, because the Authors seem to use BMI scale for adults to assess children while it should refer to other type of BMI assassment or CDC-charts (Goluch-Koniuszy ZS, Kuchlewska M. Body composition in 13-year-old adolescents with abdominal obesity, depending on the BMI value. Adv Clin Exp Med. 2017 Sep;26(6):973-979. doi: 10.17219/acem/61613. )
  8. You cannot read anything form fig. 3 - please, add more information on what it presents
  9. Refer to asymmetrical and symmetrical tooth development, especially int he discussion (you can find articles regarding Demirijans method or the charts of tooth eruption) and compare it to the asymmetrical mandibular growth BabczyÅ„ska, A.; Kawala, B.; Sarul, M. Genetic Factors That Affect Asymmetric Mandibular Growth—A Systematic Review. Symmetry 202214, 490. https://doi.org/10.3390/sym14030490
  10. After the line 201, the information weather there was a difference between sexes is missing as well as if there was any hypodontia in the examined patients (or was it taken into account?)
  11. Would you be able to explain the diference in the discussion between the Hispanic and Caucasian race and try to refer to the low number of this first group in your examination (although the region would suggest there should be more Hispanic people there)
  12. In the discussion (as well as limitations) there should be a statement on presence or lack of presence of people with dentofacial anomalies in the examined group
  13. Please, add the Comittee of Bioethics permission
  14. In lines 220-227 there should be written that the Authors have no knowledge on that kind of research - the researches had been done in the past decades, but published in the language of the country they originated (in most cases)
  15. Because of many flaws in this study, the Authors should add limitations and possible flaws made by Authors in this study

Thank you and I wish you a positive evaluation of this paper.

Author Response

                Thank you for your time and considerations. Next, we proceed to specify the changes we have made to the manuscript, which we hope will increase its quality.

              It is true that some of the references are old, since the basic anatomical and functional bases were described long ago, however we consider them important as a conceptual framework. In addition, there are not many studies that analyze the relationship between body mass index and tooth eruption, so we have had to incorporate old articles.

However, we have proceeded to incorporate some recent articles such as the following in the introduction. In accordance with its indications, we have briefly added in the introduction that dentigerous cysts and dentofacial anomalies (especially cleft palate) can alter the dental eruption process with their respective references. However, we have not proceeded to discuss it, since these patients were excluded from the study and therefore were not analyzed, we proceed to specify it more clearly in the study exclusion criteria. “Recently, mutations in the PTHR1 gene have been associated with primary eruption failure (arrest of supraosseous eruption) and mutations in GNAS with other dental eruption defects [Kurosaka]. Other factors such as dentigerous cysts and craniofacial deformities such as cleft palate have been described in association with abnormalities in dental eruption [Nahajowski et al, Paradowska-Stolarz A et al, Almotairy et al].”

  • Kurosaka H, Itoh S, Morita C, Tsujimoto T, Murata Y, Inubushi T, Yamashiro T. Development of dentition: From initiation to occlusion and related diseases. J Oral Biosci. 2022 Feb 26:S1349-0079(22)00042-1. doi: 10.1016/j.job.2022.02.005. Epub ahead of print. PMID: 35231627.
  • Nahajowski M, Hnitecka S, Antoszewska-Smith J, Rumin K, Dubowik M, Sarul M. Factors influencing an eruption of teeth associated with a dentigerous cyst: a systematic review and meta-analysis. BMC Oral Health. 2021 Apr 7;21(1):180. doi: 10.1186/s12903-021-01542-y
  • Paradowska-Stolarz A. MSX1 gene in the etiology orofacial deformities. Postepy Hig Med Dosw (Online). 2015 Dec 31;69:1499-504. PMID: 27259221.
  • Almotairy N, Pegelow M. Dental age comparison in patients born with unilateral cleft lip and palate to a control sample using Demirjian and Willems methods. Eur J Orthod. 2018 Jan 23;40(1):74-81. doi: 10.1093/ejo/cjx031

Initially, the graphics of boys and girls were not added, since they are in Spanish. They are added and referenced in the appendix.

Kilic's study (Kiliç M, Gurbuz T, Kahraman CY, Cayir A, Bilgiç A, Kurt Y. Relationship between the TAS2R38 and TAS1R2 polymorphisms and the dental status in obese children [published online as ahead of print on May 4, 2022] Dent Med Probl. doi:10.17219/dmp/143252) analyzes the relationship between obesity and caries rates, which does not correspond to the objective of our study. It is briefly mentioned in the theoretical framework of the introduction (lines 67-69).

We have not cited the article by DeszczyÅ„ska et al (DeszczyÅ„ska K, Górska R, HaÅ‚adyj A. Clinical condition of the oral cavity in overweight and obese patients. Dent Med Probl. 2021;58(2):147–154. doi:10.17219/ dmp/127873) since it is for adult patients, but we have proceeded to add the following in the introduction to broaden the conceptual framework of the oral health status in the pediatric population with obesity/overweight:

  • Schmidt J, Vogel M, Poulain T, Kiess W, Hirsch C, Ziebolz D, Haak R. Association of Oral Health Conditions in Adolescents with Social Factors and Obesity. Int J Environ Res Public Health. 2022 Mar 2;19(5):2905. doi: 10.3390/ijerph19052905. PMID: 35270598; PMCID: PMC8910061.
  • Vaziri F, Bahrololoomi Z, Savabieh Z, Sezavar K. The relationship between children's body mass index and periodontal status. J Indian Soc Periodontol. 2022 Jan-Feb;26(1):64-68. doi: 10.4103/jisp.jisp_899_20. Epub 2022 Jan 1. PMID: 35136319; PMCID: PMC8796783.
  • Iglesias Yunes EA, Costa Martins A, Feitoza de Jesus S, Juber P, Boabaid Loureiro B, Cruvinel Zuza E. Prevalence of Periodontal Disease and Alveolar Bone Loss in Overweight/Obese Brazilian Adolescents. J Dent Child (Chic). 2021 Sep 15;88(3):196-201. PMID: 34937630.
  • Traver-Ferrando C, Barcia-González J. Early permanent dental eruption in obese/overweigh schoolchildren. J Clin Exp Dent. 2022 Feb 1;14(2):e199-e204. doi: 10.4317/jced.58568. PMID: 35173904; PMCID: PMC8842294.

We proceed to add “Significant relationship was found between low weight due to malnutrition and delayed eruption of the permanent dentition and the presence of untreated caries in primary dentition in a longitudinal study in children in Indonesia” in the introduction, according to the results of the study. study by Damaisip-Nabuab et al.

Thank you for your considerations regarding BMI classification in children. The Goluch-Koniuszy study you refer to categorizes the BMI based on percentiles. In our case, we evaluated the different classification options, and finally decided to use the IOTF (International Obesity Task Force) criteria to classify childhood obesity. In addition, this classification was chosen since it is the one used in other investigations, in order to adequately carry out the comparison and discussion with previous literature.

  • Cole TJ, Lobstein T. Extended international (IOTF) body mass index cut-offs for thinness, overweight and obesity. Pediatr Obes. 2012 Aug;7(4):284-94. doi: 10.1111/j.2047-6310.2012.00064.x. Epub 2012 Jun 19. PMID: 22715120.
  • WHO: https://www.who.int/news-room/fact-sheets/detail/obesity-and-overweight

We proceed to increase the size of figure 3 and specify in the reading the information it represents.

Thank you very much for the consideration, we had not thought of this consideration. After studying the subject, we have proceeded to add to the discussion “In a recent systematic review [Babczynska] analyzed the genetic factors associated with asymmetric mandibular growth, finding potential etiological factors including PITX2, ACTN3, ENPP1 and ESR1. However, to date the scientific evidence of this association is low, and furthermore no significant relationship has been found between asymmetric mandibular growth and asymmetry in dental eruption.”

No relationship has been mentioned between the eruption sequence with respect to sexes since no significant differences were found, it is added to the corresponding paragraph. Regarding hypodontia, those children with dental agenesis were excluded (concept of dental anomaly). It is added to the exclusion criteria.

Differentiation was made between Hispanic and Caucasian to avoid risk of bias due to being two different races. We proceed to change Hispanic to South Americans, since in Spain the predominant race is Caucasian. Due to the characteristics of the study sample and the majority of the Caucasian population in Spain, the group of South American children was significantly lower.

As we have already determined, dentofacial anomalies are an exclusion criterion for our study, which is why they were not analyzed.

We proceed to attach the acceptance document by the Bioethics Committee. The acceptance number is specified in the material and methods section and at the end of the manuscript in the specific section.

The articles mentioned in the lines you mention (220-227 of the original manuscript) have been read. However, when conducting the discussion, only those similar to our study have been chosen in order to carry out the comparison between them.

A section on limitations of the study is added.

Round 2

Reviewer 1 Report

Dear Authors, 

I foud your article has been improved by corrections.

Author Response

Thank you very much for your time and your considerations. The authors believe that thanks to your corrections and suggestions, as well as those of the other reviewers, we have improved the quality of our manuscript. 

We have improved some spelling and formatting errors, eg in the abstract "southamericans children" we have changed to "southamerican children". We have corrected the same error throughout the manuscript. "The present study present" has been modified by "The present study set out" because it is redundant."

Reviewer 2 Report

Dear Sirs, thank you for a huge letter to explain all the details in the study.

Here are only 2 more tiny suggestions:

1. please, reject no. 52 from references (or is there something missing?)

2. I would add to the discussion the aspect I mentioned - about BMI and the percentile grids used for assessment of children's weight and weight-to-height ratio. 

After that, the paper could be accepted. Thank you

Author Response

Thank you very much for the corrections made, it is true that the manuscript has improved considerably. We proceed to specify the last changes made.

 Reference 52 was a tabulation error, there is no such reference, the corresponding line is removed. 

Regarding the considerations with the references and tables used for the BMI, it is proceeded to add in the discussion section in two areas. In the first paragraph (discussion of the method, line 229 above) it is added "The choice of these criteria was due to the fact that they are the most used for the classification of childhood obesity, and therefore, for the performance of an adequate comparison and discussion between articles. Thus, since the sample was of Spanish children, it was decided to use BMI tables in the Spanish population so that they would be representative." 

In the last paragraph of the discussion (limitations of the study, line 338 above) "Since the values and classification criteria of the BMI are variable, it is important to bear in mind that those used for the present study correspond to those in force in the Spanish population and at the date of publication, so caution should be exercised when interpreting these data in other populations or races."